# Inverse Design of Valley-Like Edge States of Sound Degenerated Away from the High-Symmetry Points in a Square Lattice

**DOI:** 10.3390/ma15196697

**Published:** 2022-09-27

**Authors:** Jishi Yang, Yaolu Liu, Dongyang Sun, Ning Hu, Huiming Ning

**Affiliations:** 1College of Aerospace Engineering, Chongqing University, Chongqing 400044, China; 2Nanjing Research Institute, Chongqing University, Nanjing 211800, China; 3School of Mechanical Engineering, and National Engineering Research Center for Technological Innovation Method and Tool, Hebei University of Technology, Tianjin 300401, China

**Keywords:** inverse design, valley-like states, edge states, explicit geometry description

## Abstract

Robust edge states of periodic crystals with Dirac points fixed at the corners or centers of the Brillouin zones have drawn extensive attention. Recently, researchers have observed a special edge state associated with Dirac cones degenerated at the high symmetric boundaries of the first irreducible Brillouin zone. These nodal points, characterized by vortex structures in the momentum space, are attributed to the unavailable band crossing protected by mirror symmetry. By breaking the time reversal symmetry with intuitive rotations, valley-like states can be observed in a pair of inequivalent insulators. In this paper, an improved direct inverse design method is first applied to realize the valley-like states. Compared with the conventional strategy, the preparation of transition structures with degeneracy points is skipped. By introducing the quantitative gauge of mode inversion error, insulator pairs are directly obtained without manually tuning the structure with Dirac cone features.

## 1. Introduction

Topological materials have drawn wide attention in the fields of condensed matter physics and engineering [1,2,3,4]. Valley materials are first proposed in graphene structures [5,6,7,8]. Similar two-dimensional materials are observed in other hexagonal lattices [9,10,11,12,13,14,15,16,17], such as bilayer graphene [15,16,17] and transition metal dichalcogenides [13,14], which are typical valleytronic materials.

Similar to electron spin, topological valley transport, which is usually induced by breaking the inversion symmetry of the system, has the potential to transfer information and energy. Inspired by the intriguing ability of supplying a topologically protected transport with high transmission rate and good robustness against backscattering, tremendous progress has been made in the study of non-trivial states in different physical systems, e.g., electromagnetic [18,19,20,21,22,23,24], acoustics [25,26,27,28,29,30,31,32,33,34] and mechanics [35,36,37,38,39,40,41].

Current studies mainly focus on the Dirac cones [42] or double Dirac cones [43,44,45] fixed at the high-symmetric points of the Brillouin zones in hexagonal lattices. However, phononic insulators beyond typical graphene-like lattices are relatively rare. Existing studies only focus on the realization of valley states or valley-like states through conventional procedures by: (1) finding a structure with Dirac cones; (2) splitting the Dirac cone by introducing a disturbance, which breaks the inversion symmetry or mirror symmetry of the base structure; (3) verifying the physical properties of the outcomes after disturbance. From a practical point of view, the desired wild bandgap (i.e., operation bandwidth) may not be obtained through the intuitive procedure mentioned above. Due to the existence of the base structure, the design space for breaking the symmetries is very limited. In general, designers can only choose from some common operations, such as rotation [28], offset [46], and zooming [47]. The trial-and-error procedure lacks mathematical optimality and is often time consuming. To overcome the aforementioned drawbacks, inverse design methods have been proposed to design topological materials with Dirac cones [48,49]. Furthermore, topological insulator pairs have been successfully obtained at certain frequencies by optimizing the transmission path [50,51] or the field distribution of eigenmodes [52,53].

In this paper, a direct inverse design method is adopted to realize valley-like edge states. The geometries of potential insulators are described by the B-spline method. By setting control points in an equiangular distribution, an arbitrary geometry can be determined by a finite set of parameters. In addition, by introducing symmetry restrictions, mirror symmetry breaking and control parameter reduction are achieved simultaneously, which are beneficial to gradient-free algorithms. By utilizing band inversion, mode inversion error is set as a quantitative gauge that is used as a constraint in the design process. Additionally, maximizing the width of common bandgap is set as the objective function. Then, the inverse design is converted into an optimization problem, which can be solved by a genetic algorithm. Compared with the design in the literature [34], the relative bandgap of the inverse design is significantly improved by around 70%.

## 2. Description of the Geometry of Scatters

According to existing studies [34,54], valley-like states can be realized with only one scatter in a square lattice. Thus, it is appropriate to describe the geometry of scatter with a B-spline curve. As shown in Figure 1, an arbitrary geometry can be described with well-constructed control points. Through introducing extra constraints on the distance between the control point and the origin point, different symmetries can be effectively realized. Moreover, a reference angle is introduced to improve the representation of scatters. After many trials, we found that 14 control points were sufficient for the geometry description, which can be described by a B-spline formulated as:
(1)$$C(u)=\sum_{i=0}^{n+1}N_{i,2}(u)P_{i},(u_{lower}\leq u\leq u_{upper})$$
where *n* is the number of independent control points (i.e., *n* = 12) and *P_i_* represents the coordinates of control points, which can be expressed as:(2)$$P_{i}={\left(r_{i}\cos((i-1)\frac{\pi }{6}+\theta ),r_{i}\sin((i-1)\frac{\pi }{6}+\theta )\right)}^{T},i=1,\ldots ,n$$
with $r_{1},\ldots ,r_{n}$ denoting the distances from the origin point to the control points and θ denoting the reference angle between the *x*-axis and the line along the origin point and $P_{1}$. In order to have a closed B-spline curve, two independent control points are introduced, i.e., $P_{0}=P_{12}$ and $P_{13}=P_{1}$. By merging the first two control points with the last two control points, the continuity of the closed curve is guaranteed. In addition, the second order B-spline basis function $N_{i,2}$ can be obtained with recurrence formulas as follows:(3)$$N_{i,0}(u)=\left\{\begin{array}{ll} 1,\; & {\;if\;u}_{i}\leq u<u_{i+1}\\ 0,\; & otherwise \end{array}\right.$$
(4)$$N_{i,p}(u)=\frac{u-u_{i}}{u_{i+p}-u_{i}}N_{i,p-1}(u)+\frac{u_{i+p+1}-u}{u_{i+p+1}-u_{i+1}}N_{i+1,p-1}(u)$$

Since the first *p* and last *p* control points are wrapped together (*p* = 2), the value of knot $u_{i}$ and the numerical range of parameter u in Equation (1) should be in the following form: (5)$$\begin{array}{l} u_{i}=\frac{i}{n+p+1},\;\;\;\;\;\;\;\;\;\;\;i=0,1,\ldots ,n+p+1\\ u_{lower}=\frac{p}{n+p+1}\\ u_{upper}=\frac{n+1}{n+p+1} \end{array}$$

Then, the closed B-spline curve can be determined by a vector $V={\left(r_{1},r_{2},\ldots ,r_{12},\theta \right)}^{T}$ accurately. By flexibly setting these parameters, the group symmetries can be inherently realized. For example, if we set $r_{i}=r_{i+3}=r_{i+6}=r_{i+9}(i=1,2,3)$ and θ=0, the geometry of scatters will have C4v point-group symmetry. Furthermore, a previous study has shown that the emergency of Dirac points used in this paper is only protected by mirror symmetry [55]. Thus, mirror symmetry breaking can be easily realized by introducing C4 symmetry in the geometry of scatters when we set $r_{i}=r_{i+3}=r_{i+6}=r_{i+9}(i=1,2,3)$. Inspired by the reference design [34], chiral constraints can also be introduced in our design procedure to describe a pair of insulators and reduce the design variables. Then, the number of independent variables is reduced to 4, i.e., $V={\left(r_{1},r_{2},r_{3},\theta \right)}^{T}$. The calculation cost of gradient-free algorithms such as the genetic algorithm is acceptable when applied to these highly reduced design variables.

## 3. Inverse Design

### 3.1. Quantitative Measures of Band Inversion

Studies have shown [28,48] that the emergency of localized topological states is accompanied by band inversion, i.e., parts of the relevant two bands will exchange when nontrivial states occur. In addition, the relevant mode fields in a phononic crystal with different chiralities will be reversed compared with the other crystals. The quality of band inversion can be quantified by the difference of the modes of two chiral phononic crystals around Dirac cones. To achieve such requirements, the mode inversion error of two chiral crystals about their *i*-th and (*i* + 1)-th bands is introduced as
(6)$$E_{i,i+1}=\frac{\int {}_{\Omega }(\left|{\bar{\left|\psi \right|}}_{A,i}-{\bar{\left|\psi \right|}}_{B,i+1}\right|+\left|{\bar{\left|\psi \right|}}_{B,i}-{\bar{\left|\psi \right|}}_{A,i+1}\right|)d\Omega }{\int {}_{\Omega }(\max({\bar{\left|\psi \right|}}_{A,i},{\bar{\left|\psi \right|}}_{B,i+1})+\max({\bar{\left|\psi \right|}}_{B,i},{\bar{\left|\psi \right|}}_{A,i+1}))d\Omega }$$
where Ω denotes the intersection area of the solid part of these two crystals, subscript A (B) represents the chiral crystal A (B), and $\bar{\psi }$ is the normalized amplitude field defined by the orthonormal property of the basis function [56], i.e., $\frac{\left(2\pi \right)}{\Omega }\int {}_{cell}\psi_{i,\vec{k_{0}}}^{*}(\vec{r})\frac{1}{B_{r}(\vec{r})}\psi_{j,\vec{k_{0}}}(\vec{r})d\vec{r}=\delta_{ij}$. Considering the difference between two chiral crystals and the computational accuracy, the mode inversion error should be very small but never drop to zero. It is therefore recommended to use this error as a constraint or multi-objective function during optimization. 

### 3.2. Common Bandgap Width

Considering the edge states appeared in the nontrivial common bandgap of a pair of insulators, we adopt the common bandgap width as an objective function, which is formulated as:(7)$$G_{i,i+1}=\min((f_{A,i+1}-f_{A,i}),(f_{B,i+1}-f_{B,i}),(f_{A,i+1}-f_{B,i}),(f_{B,i+1}-f_{A,i}))$$
where subscript $f_{A,i}$ represents the frequency of the *i*-th band of the crystal A. It should be noted that, though the value of *i* used in this paper is four, the value can be different depending on specific situations. For example, parameter *i* can be six in a 2D photonic crystal with a triangular lattice [55] or one in a 2D sonic crystals with a triangular lattice [28]. 

### 3.3. Problem Formulations

Inverse design of valley-like states will be achieved by applying the objective optimization in the following form:
(8)$$\begin{array}{l} For\;k\;on\;XM:\\ Find:\;V={\left(r1,r2,r3,\theta \right)}^{T}\\ Maximize:\;G_{i,i+1}=\min((f_{A,i+1}-f_{A,i}),(f_{B,i+1}-f_{B,i}),(f_{A,i+1}-f_{B,i}),(f_{B,i+1}-f_{A,i}))\\ Subject\;to:\;\nabla \cdot \left[\frac{1}{\rho_{r}(\vec{r})}\nabla p\right]=-\frac{\omega^{2}}{c^{2}}\cdot \frac{p}{B_{r}(\vec{r})}\\ \;\;\;\;\;\;\;\;\;\;\;\;\;\;\;\;\;\;\;\;\;\;E_{i,i+1}=\frac{\int {}_{\Omega }(\left|{\bar{\left|\psi \right|}}_{A,i}-{\bar{\left|\psi \right|}}_{B,i+1}\right|+\left|{\bar{\left|\psi \right|}}_{B,i}-{\bar{\left|\psi \right|}}_{A,i+1}\right|)d\Omega }{\int {}_{\Omega }(\max({\bar{\left|\psi \right|}}_{A,i},{\bar{\left|\psi \right|}}_{B,i+1})+\max({\bar{\left|\psi \right|}}_{B,i},{\bar{\left|\psi \right|}}_{A,i+1}))d\Omega }\leq E_{preset}\\ \;\;\;\;\;\;\;\;\;\;\;\;\;\;\;\;\;\;\;\;\;\;(0.1\times a)\leq r_{i}\leq (0.4\times a),i=1,2,3\\ \;\;\;\;\;\;\;\;\;\;\;\;\;\;\;\;\;\;\;\;\;\;0\leq \theta \leq \frac{\pi }{6} \end{array}$$
where a represents the lattice constant (a=24mm); the density and the sound speed of wave medium (air) are 1.25 kg/m^3^ and 343 m/s; $\rho_{r}(\vec{r})$ and $B_{r}(\vec{r})$ are the relative mass density and bulk modulus, respectively; $E_{preset}$ is a positive number, which is close to zero; $r_{i}$ represents the distance from the origin point to the control point $P_{i}$. Additionally, the value of $r_{i}$ is restricted to [0.1×a,0.4×a] in order to avoid self-interpretation.

## 4. Valley-Like Vortex States

The geometry of phononic crystals has C4 point-group symmetry, which leads to the mismatch in mirror symmetries between the square scatter and lattice. Since our method is a direct method, the construction procedure of the specific Dirac cones is skipped and the transition structure remains unknown. The outcomes of the imaginary structure after a so-called open–close–reopen operation [28,48] are illustrated in Figure 2, which clearly shows that the eigenmodes of the fourth and fifth bands are inversed near the Dirac cones, i.e., a valley Hall phase transition is induced. This phenomenon can be regarded as a band inversion process, which is similar to the valley Hall phase transition in electronic systems. 

Moreover, the acoustic valley-like states possess chiral characteristics. Figure 2 shows the notable features of vortices of power flows in these two different crystals. The forth eigenmode of crystal A exhibits the same vortex chirality as the fifth eigenmode of crystal B. The two eigenmodes of crystal A (or B) exhibit the opposite vortex chirality, while the forth eigenmode of crystal B shows the same vortex chirality with the fifth eigenmode of crystal A. In other words, the inversion of the acoustic valley-like states results in an analogy of the valley Hall phase inversion.

## 5. Valley-Like Edge States

For a supercell consisting of a 20 × 1 unit cell with rigid boundaries at the top and bottom and Bloch periodic conditions at the left and right boundaries, the array of the unit cells can be regarded as a quasi-one-dimensional structure. Through adjusting the configuration of crystals and comparing relevant energy bands, the edge bands will be observed efficiently. As shown in Figure 3, the supercell with 20-unit cells of the same structure (crystal A) can be regarded as a trivial material, and the two overlapped edge bands emerge inside the bulk bandgap. By observing the mode fields, we can find that one is the top edge mode while the other is the bottom edge mode. No interface mode occurs in this trivial configuration. 

We design a new supercell containing 20 × 1-unit cells with different structures, i.e., the upper 10 × 1-unit cells consist of crystal A while the lower 10 × 1-unit cells consist of crystal B. Different crystals at the opposite sides of the interface show different valley Hall phases. Four edge states can be observed in the bulk bandgap, as shown in Figure 1. These edge bands represent four different edge modes. Two of these modes are edge modes located at the top and bottom boundary which are similar to those of the supercell consisting of the same unit cells, while the other two modes are the interface modes. The acoustic valley-like edge states accompanied with clockwise and counterclockwise vortices can robustly lead to the propagation of energy flow in opposite directions. 

## 6. Observation of Energy Propagation against Defects

Figure 4a shows a perfect structure consisting of a pair of valley-like insulators. Two different defects, i.e., cavities and disorders, are considered in Figure 4b,c, which are very common during the processing procedure. The cavity defects are induced by removing several unit cells at the interface between two different structures. Compared with that in Figure 4e, the disturbance can seriously affect the propagation ability of the band-gap-guiding transmission consisting of trivial insulators, but has little influence on the nontrivial valley-like design. The second type of defects is the disorders waveguide, which is induced by the random rotated scatters along the path of wave propagation. The amplitude of rotation is 20% of the upper limit of the reference angle, i.e., the design parameter θ in Equation (2). For any trivial guided edge mode shown in Figure 4, these defects can cause strong backscattering and even block wave propagation. Since cavities and disorders are not spin-mixing defects, the valley-like edge states along the interface will not be broken by these “nonmagnetic” impurities [57,58]. The simulated acoustic pressure fields shown in Figure 4b,c indicate that the wave incident from the left side can bypass all these defects at the valley-like protected interface and then efficiently propagate to the right side without significant attenuation. The valley-like protection has suppressed the backscattering of acoustic waves and ensured the robust transmission against various types of defects, including cavities and disorders.

As shown in Figure 5, we also investigate the transmission properties of the valley-like edge states. The black line in Figure 5b represents the transmission spectrum of the perfect configuration with no defect. The two typical defects mentioned in Figure 4b,c are represented by the red and blue lines, respectively. As a comparison, the transmission spectrum of a trivial structure formed by a single crystal (crystal A) is given in Figure 5b (the green line). The excitation sources are the same for all of these cases. By comparison, the transition of the trivial structure in the bulk bandgap is very low. In contrast, the valley-like insulators with a perfect configuration exhibit very high transmission. Moreover, high-transmission spectra can be observed in two imperfect interfaces with defects, suggesting that our valley-like structure has robust edge transmission properties against “non-magnetic” defects. 

## 7. Conclusions

In conclusion, this paper shows an inverse design method for valley-like edge states. By quantifying the mode inversion error and maximizing the common bandgap, the inverse design can be transformed into an optimization problem. Instead of constructing Dirac cones at the momentum away from the high-symmetric points of the square lattice, the products are directly obtained after the break of mirror symmetries. Such a design has robust transmission properties against defects and can guide sound waves through the twisted path despite suffering two sharp corners (bent by 135° as shown in Figure 6). Compared to the literature [34], this method does not require trial-and-error design and achieves a larger operation bandgap (70% wider than the original design). Since the inverse strategy does not require certain physical fields, it can theoretically be applied to other classical systems such as optical systems and mechanical systems.

## Figures and Tables

**Figure 1 materials-15-06697-f001:**
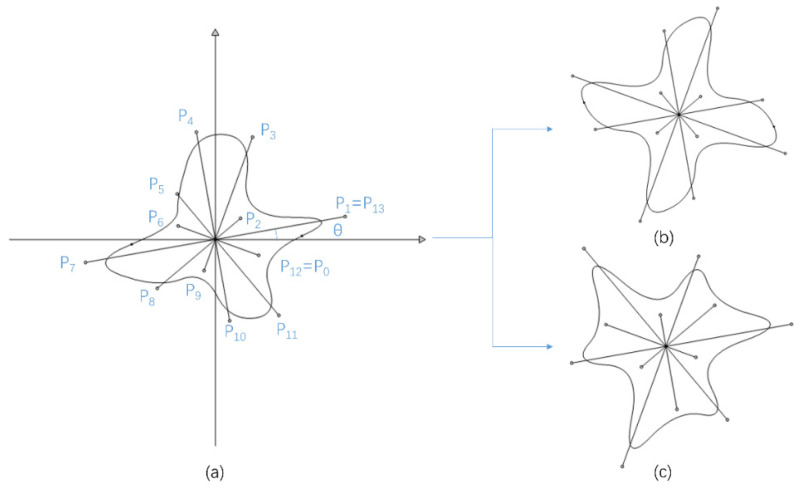
(**a**) An illustration of the potential scatter described by a B-spline curve with 14 control points. (**b**) A C_4_-symmetric design with reduced independent parameters, i.e., $r_{i}=r_{i+3}=r_{i+6}=r_{i+9}(i=1,2,3)$. (**c**) A C3-symmetric design with constraints $r_{i}=r_{i+4}=r_{i+8}(i=1,2,3,4)$.

**Figure 2 materials-15-06697-f002:**
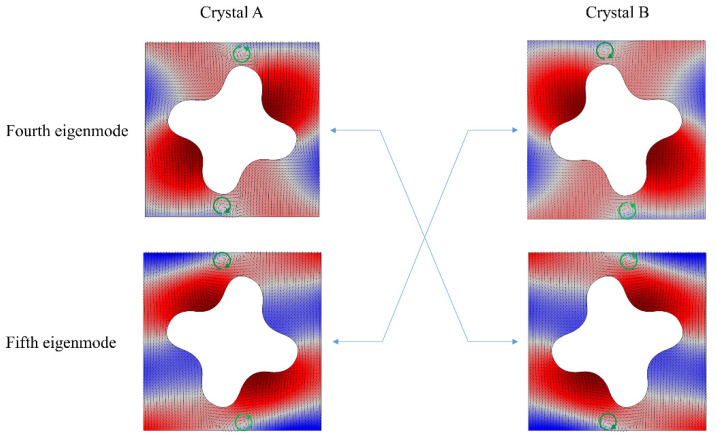
An illustration of band inversion of a pair of valley-like insulators after the break of mirror symmetries near the original Dirac point. Accompanied with the reopen of the degenerate point, the fourth eigenmode of crystal A is exchanged with the fifth eigenmode of crystal B. The typical vortex feature can be observed in the time-averaged Poynting vectors marked with additional green arrows.

**Figure 3 materials-15-06697-f003:**
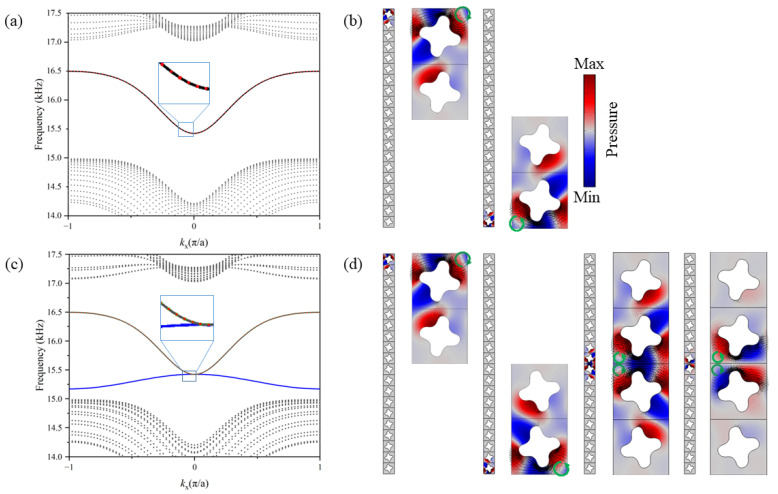
(**a**) The energy band of a supercell consisting of 20 × 1-unit cells with the same structure (crystal A). (**b**) The eigenmodes (acoustic pressure field) localized at the top boundary with clockwise vortex and at the bottom with counterclockwise vortex. (**c**) The energy band of a supercell consisting of 20 × 1-unit cells with different structures (crystal A and crystal B). (**d**) The acoustic pressure field shows the edge states localized at the top boundary and the bottom boundary, as well as the interface states between the two different crystal structures. In (**a**,**c**), the red dotted line and the black solid line represent the edge states localized at the top boundary and the bottom boundary, respectively; the green hollow circle and the blue solid line indicate the two interface states induced by the valley-like insulators; the gray dotted lines represent the bulk bands. In (**b**,**d**), the arrows in the acoustic eigenmode fields represent the Poynting vectors; the additional green arrows indicate the location of vortex cores.

**Figure 4 materials-15-06697-f004:**
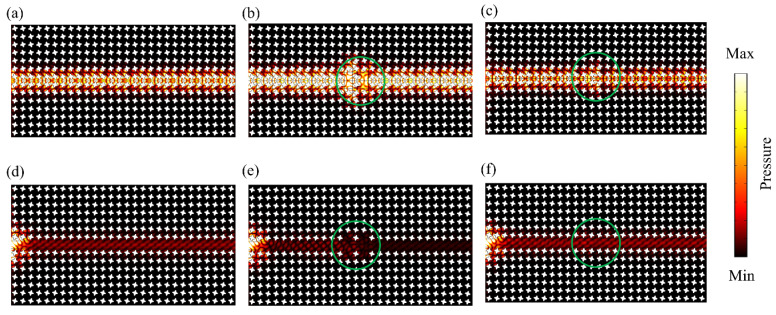
(**a**) Simulated acoustic pressure field in the valley-like structures without defects. (**b**,**c**) Simulated acoustic pressure field with cavities and disorders, where the incident frequency of the acoustic wave is 16 kHz (within the bulk bandgap). (**d**) Simulated acoustic pressure field of the trivial band-gap-guiding structures without defects. (**e**,**f**) Simulated acoustic pressure field with cavities and disorders. The 4 green circles in subfigure (**b**,**c**,**e**,**f**) mark the location of defects.

**Figure 5 materials-15-06697-f005:**
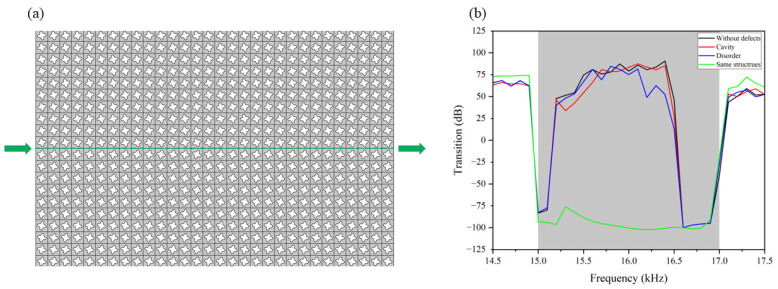
(**a**) Schematic diagram for the investigation of transmission properties. (**b**) Transmission spectra of different structures. The green line represents the trivial design consisting of only one type of insulator (crystal A). The black, red, and blue curves represent the structures consisting of a pair of different insulators with no defects, with cavities and with disorders, respectively.

**Figure 6 materials-15-06697-f006:**
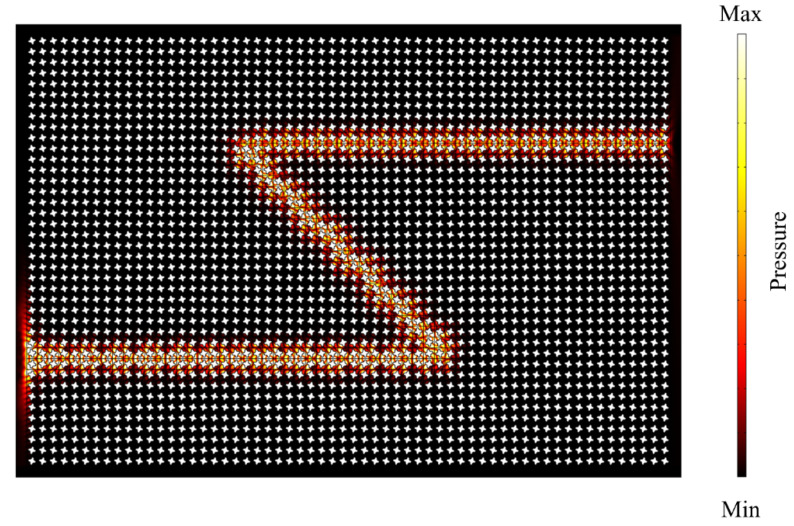
Simulated acoustic pressure field along a curved path with sharp corners (bent by 135°).

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
