# Peer review of "Inverse Design of Valley-Like Edge States of Sound Degenerated Away from the High-Symmetry Points in a Square Lattice"

_materials, 2022, doi:10.3390/ma15196697_

Round 1

Reviewer 1 Report

In this manuscript, the authors have shown an inverse design method for valley-like edge states. Instead of constructing Dirac cones at the momentum away from the high-symmetric points, the products are directly obtained after breaking mirror symmetries. They show their proposed design had robust transmission properties against defects, and their method does not require trial-and-error design and achieves a larger operation bandgap. I recommend that this manuscript for published, but I have some suggestions to improve the quality of the paper.

 1.     From where did the authors get the inspiration for this specific design?

 2.     Authors should include the possibility of fabrication and experimentation in their proposed work.

 3.     The authors claimed that their method does not require trial-and-error design and achieves a larger operation bandgap. Is this specific for square lattice, or can it be applied to any other proposed structure like hexagonal lattice? 

Round 2

Reviewer 2 Report

The authors have addressed all my questions. I recommend the article for publication.